# Complex Karyotype Detection in Chronic Lymphocytic Leukemia: A Comparison of Parallel Cytogenetic Cultures Using TPA and IL2+DSP30 from a Single Center

**DOI:** 10.3390/cancers16122258

**Published:** 2024-06-18

**Authors:** Joanna Kamaso, Anna Puiggros, Marta Salido, Carme Melero, María Rodríguez-Rivera, Eva Gimeno, Laia Martínez, Leonor Arenillas, Xavier Calvo, David Román, Eugènia Abella, Silvia Ramos-Campoy, Marta Lorenzo, Ana Ferrer, Rosa Collado, Marco Antonio Moro-García, Blanca Espinet

**Affiliations:** 1Molecular Cytogenetics and Hematological Cytology Laboratories, Pathology Department, Hospital del Mar, 08003 Barcelona, Spain; jkamaso@researchmar.net (J.K.); msalido@psmar.cat (M.S.); mmelero@psmar.cat (C.M.); mmrodriguezrivera@psmar.cat (M.R.-R.); larenillas@psmar.cat (L.A.); xcalvo@psmar.cat (X.C.); oroman@psmar.cat (D.R.); sramos@researchmar.net (S.R.-C.); mlorenzo@psmar.cat (M.L.); aferrera@psmar.cat (A.F.); 2Translational Research on Hematological Neoplasms Group, Cancer Research Program, Hospital del Mar Research Institute (HMRI), 08003 Barcelona, Spain; 3Department of Hematology, Hospital del Mar, 08003 Barcelona, Spain; egimenov@psmar.cat (E.G.); mabella@psmar.cat (E.A.); 4Applied Clinical Research in Hematological Malignances Group, Cancer Research Program, Hospital del Mar Research Institute (HMRI), 08003 Barcelona, Spain; 5Hematology Service, Hospital Universitari Sant Joan de Reus, 43204 Reus, Spain; laia.mserra@gmail.com; 6Department of Hematology, Consorcio Hospital General Universitario Valencia, 46014 Valencia, Spain; collado_ros@gva.es; 7Laboratory Medicine Department, Hospital Universitario Central de Asturias, 33011 Oviedo, Spain; marcomorog@hotmail.com

**Keywords:** (3–10): mitogen, TPA, IL2, DSP30, chronic lymphocytic leukemia, chromosome banding analysis, complex karyotype

## Abstract

**Simple Summary:**

Current recommendations suggest setting two parallel cytogenetic cultures with 12-O-tetradecanoly-phorpol-13-acetate (TPA) and IL2+DSP30 as a mitogen to detect complex karyotypes (CKs) in chronic lymphocytic leukemia (CLL). However, studies comparing CK detection concordance between both methods in the same cohort are lacking. Herein, we evaluated the performance of two parallel cultures in a CLL cohort of 255 patients, specifically comparing CK detection (globally and in each individual patient). The CK detection rates and their prognostic impacts were similar for both mitogens. However, nearly one-third of CKs were only identified in one culture, mainly due to the detection of a normal karyotype or no metaphases in the other. In summary, the assessment of parallel cytogenetic cultures is the best strategy to detect CKs in CLL. Nonetheless, as IL2+DSP30 achieved the best performance, it should be prioritized above TPA if a single analysis is required to optimize cytogenetic assessment in routine practice.

**Abstract:**

Current CLL guidelines recommend a two parallel cultures assessment using TPA and IL2+DSP30 mitogens for complex karyotype (CK) detection. Studies comparing both mitogens for CK identification in the same cohort are lacking. We analyzed the global performance, CK detection, and concordance in the complexity assessment of two cytogenetic cultures from 255 CLL patients. IL2+DSP30 identified more altered karyotypes than TPA (50 vs. 39%, *p* = 0.031). Moreover, in 71% of those abnormal by both, IL2+DSP30 identified more abnormalities and/or abnormal metaphases. CK detection was similar for TPA and IL2+DSP30 (10% vs. 11%). However, 11/33 CKs (33%) were discordant, mainly due to the detection of a normal karyotype or no metaphases in the other culture. Patients requiring treatment within 12 months after sampling (active CLL) displayed significantly more CKs than those showing a stable disease (55% vs. 12%, *p* < 0.001). Disease status did not impact cultures’ concordance (κ index: 0.735 and 0.754 for stable and active). Although CK was associated with shorter time to first treatment (TTFT) using both methods, IL2+DSP30 displayed better accuracy than TPA for predicting TTFT (C-index: 0.605 vs. 0.580, respectively). In summary, the analysis of two parallel cultures is the best option to detect CKs in CLL. Nonetheless, IL2+DSP30 could be prioritized above TPA to optimize cytogenetic assessment in clinical practice.

## 1. Introduction

Chronic lymphocytic leukemia (CLL) is the most common leukemia among adults in Western countries. The clinical course is very heterogeneous, with some patients showing an indolent course and will never require treatment, while others present with an aggressive and rapidly progressing disease that will need therapy. This heterogeneity has important consequences in clinical approaches, treatment selection, and patient outcomes [1].

Cytogenetic abnormalities in CLL are used for diagnosis, prognosis, and treatment. It is known that some of these cytogenetic aberrations represent an independent predictor of prognosis in CLL [2]. The four cytogenetic abnormalities described by Döhner et al. are the deletion in the long arm of chromosome 13 (del (13q)) as a sole abnormality, trisomy of chromosome 12, deletion in the long arm of chromosome 11 (del (11q)), and deletion in the short arm of chromosome 17 (del (17p)). To detect these genetic abnormalities, fluorescence in situ hybridization (FISH) is used as the gold standard method, and it has a performance of >80% of all CLL cases [3]. Moreover, in recent years, the identification of a complex karyotype (CK) using chromosome banding analysis (CBA) has been postulated as a prognostic and potentially predictive biomarker. The last 2018 iwCLL guidelines introduced the performance of CBA as a “desirable” test in the context of clinical trials, and more recent recommendations, such as current German CLL guidelines, are starting to stratify patients based on the presence of complex karyotype [3,4]. In addition, the optimization of CK detection in routine diagnosis will provide real-world data to elucidate its impact and, ultimately, will allow for the development of novel tailored therapeutic strategies (e.g., identifying those patients who will benefit from continuous treatments or fixed-duration therapy).

Classically, CK is defined by the presence of ≥3 abnormalities in the same clone detected by CBA. However, several studies have shown that this group is heterogeneous and defined as a high-CK group, with the presence of ≥5 abnormalities, being associated with dismal clinical outcomes, independently of the other biomarkers [5,6]. Even though it is well known that CK confers a poor response in patients treated with chemoimmunotherapy, the impact of CK is still controversial in patients who receive novel targeted agents such as Bruton tyrosine kinase inhibitors (BTKi) or BCL2-inhibitors [7,8].

The European Research Initiative on CLL (ERIC) group has published some recommendations for CK detection, in which they suggest performing CBA as a standard assessment. Notably, the detection of cytogenetic abnormalities in CLL is often hampered by the low mitotic rate of B lymphocytes in vitro, resulting in missing the abnormal CLL clone in a significant fraction of cases [9]. Historically, 12-O-tetradecanoylphobol-13-acetate (TPA) has been used in routine cytogenetics laboratories as a mitogen to increase the mitotic rate of CLL cells in order to perform CBA. However, its efficiency is restricted, revealing abnormal karyotypes in 40–50% of cases [10]. More recently, the combination of DSP30 CpG oligonucleotides plus interleukin-2 (IL2) has been shown to increase the detection rate of aberrant karyotypes in up to 80% of cases [11,12]. For that reason, the ERIC group recommends setting up two parallel cultures with different cell mitogens for each patient, one with TPA and the other with IL2 plus DSP30. Both mitogens activate the CLL cells in different ways; TPA activates the protein kinase C (PKC) in B cells and induces the proliferation of B lymphocytes. On the other hand, IL2 enhances proliferation and immunoglobulin production in B cells. In addition, DSP30 oligonucleotides increase the proliferation of CLL cells by inducing the expression of high-affinity IL2 receptors at higher levels than in normal B cells [13]. It is remarkable that current recommendations are mainly based on initial studies that assessed independent CLL cohorts by each mitogen [13,14]. To the best of our knowledge, only three publications have demonstrated the increased efficiency of the mitogen IL2+DSP30, performing two parallel cultures in the same patients [11,15,16]. Furthermore, it is important to consider that none of them specifically addressed the concordance in detecting CK between both methods. 

For this reason, we aimed to evaluate the performance of two parallel cultures using TPA and IL2+DSP30 mitogens in a CLL cohort of consecutively received unselected patients in a single center. This comparison was specially focused on CK and high-CK detection and their concordance in the complexity assessment of each individual patient. Moreover, patients who showed stable disease (not treated within 12 months after sampling) and those who required treatment within 12 months after sampling (active CLL) were also analyzed separately in order to ascertain the impact of disease status on the outcome of CBA using both mitogens.

## 2. Materials and Methods

### 2.1. Patient Cohort

A total of 255 patients with CLL (n = 175; 68.6%) and monoclonal B cell lymphocytosis (n = 80; 31.4%) were included from Hospital del Mar (n = 212), Hospital Universitari Sant Joan de Reus (n = 33), Hospital General Universitario Valencia (n = 7), and Hospital Universitario Central de Asturias (n = 3) between April 2016 and July 2022. All cases were referred to our center and were analyzed using CBA. The clinical and biological characteristics of this unselected CLL cohort are shown in Table 1. Patients were classified according to their disease status at recruitment, and those patients who received treatment within a year after the CBA study were considered active. Overall, 210 patients with stable disease and 45 patients with active disease were assessed. 

### 2.2. Chromosome Banding and Interphase FISH Analyses

For each patient, two parallel peripheral blood cultures were performed according to the standard protocols, one with TPA and the other with IL2+DSP30 [9]. Karyotypes were described following the International System for Human Cytogenetic Nomenclature (ISCN 2020) [17]. A minimum of 20 metaphases were analyzed when possible. Chromosome abnormalities were considered clonal when detected in two or more metaphases or in at least three metaphases for whole chromosome losses. In addition, abnormalities detected in a single metaphase that were confirmed by FISH or in a subsequent CBA were also considered. Culture failure was defined as less than 10 metaphases found in the culture. CK was defined as the presence of three or more chromosomal abnormalities in a single clone. For complexity analyses, the CKs identified were further subdivided into low/intermediate-CK (3–4 abnormalities) and high-CK (≥5 abnormalities) [6]. For the purpose of this study, comparative analyses considered the results obtained from each culture independently. Nonetheless, in order to define the clinical and biological characteristics associated with the complexity or the agreement between methods, those patients with a CK in at least one of the cultures were included in a global CK group. 

FISH was performed on uncultured fixed peripheral blood samples obtained at the same time as sampling for CBA cultures. All 255 cases had successful results. The FISH panel included four probes to investigate loci commonly involved in CLL as follows: *TP53* (17p13), *ATM* (11q22), D13S319 (13q14), and the chromosome 12 centromeric region (Metasystems, Altlussheim, Germany). A minimum of 100 nuclei were analyzed for each FISH probe, and the following cut-offs for positivity were used: 5% for 13q deletion, 3% for trisomy 12, 4% for ATM deletion, and 9% for TP53 deletion. 

### 2.3. Comparison between TPA and IL2+DSP30 Cultures

Firstly, the performance in detecting abnormalities using each mitogen on its own and globally was evaluated. In addition to abnormality and CK detection, the number and type of abnormalities detected were compared for the whole cohort and in each individual patient. Secondly, two groups according to disease status were analyzed separately to determine the impact on the concordance regarding CK detection between cultures. Finally, in order to determine the potential impact on the prognostic value of CK due to the discrepancies found between methods, the accuracy of CK detected by each mitogen in predicting time to first treatment (TTFT) was compared.

### 2.4. Statistical Analysis

Descriptive statistics were used to provide frequency distributions of discrete variables, while statistical measures were used to provide median values and ranges for quantitative variables. The different groups established were compared with chi-square or Fisher exact tests for discrete variables and the Mann–Whitney U test for continuous variables. The concordance in patient classification based on complexity between TPA and IL2+DSP30 mitogens was established using the Kappa coefficient.

Regarding survival analysis, only those 234 patients who were treatment-naïve at the time of CBA were used. TTFT was calculated from the date of the cytogenetic study to the date of first treatment or last follow-up. The Kaplan–Meier method was used to estimate the distribution of TTFT depending on the number of alterations identified by CBA. Comparisons among patient subgroups were performed via the log-rank test. The concordance statistic (C-index) was calculated to assess the accuracy in predicting TTFT. Statistical analyses were performed using SPSS v.23 software (SPSS Inc., Chicago, IL, USA). *p*-values < 0.05 were considered statistically significant.

## 3. Results

### 3.1. Global Performance of the Mitogens

In terms of the global performance of the mitogens, the karyotype could be analyzed in 236 cases after TPA culture and in 243 after IL2+DSP30 culture (success rate 93% and 95%, respectively, *p* = 0.265). When results from both cultures were combined, the success rate increased up to 99%, and only two cases remained uninformative. Regarding the detection of abnormalities, altered karyotypes were identified in 39% of TPA-based cultures (n = 100) and 50% of cultures with an IL2+DSP30 combination (n = 127). Despite the use of IL2+DSP30, mitogens achieved a significantly higher detection of abnormal karyotypes (*p* = 0.031), and the overall aberration detection increased to 53% (n = 136) when both methods were combined. While the vast majority of altered cases were detected using both methods (n = 91), 9 and 36 patients showed abnormalities using only TPA and IL2+DSP30, respectively (Table 2). Of note, among those patients with abnormalities found in both cultures, IL2+DP30 performed better than TPA in 71% of cases; more altered metaphases and/or more cytogenetic alterations were detected (Figure 1). In addition, unrelated abnormalities were found in the parallel karyotypes in two patients. The first showed a complex translocation involving t (3;14) using TPA and the loss of chromosome X using IL2+DSP30, which has been associated with advanced age rather than a clonal aberration due to CLL. The second showed two populations of clonal B cells (42% of B cells showing a phenotype consistent with splenic marginal zone lymphoma and 5% with a typical CLL phenotype) that could be differentially expanded by each mitogen. Thus, TPA stimulation allowed for the detection of an abnormal karyotype with a del(7) (q32q35), frequently described in marginal lymphomas, and trisomy 12 was observed in all the metaphases assessed in the IL2+DSP30 culture. On the other hand, IL2+DSP30 allowed for identifying abnormalities in 20% (27/136) of the patients with normal karyotypes found using TPA. In contrast, aberrations found using TPA could only be found in 4% (5/116) of the non-altered cases using IL2+DSP30 (Table 2). 

### 3.2. Chromosomal Aberrations Detected

Globally, the total number of aberrations recorded in the IL2+DPS30 karyotypes was higher than those recorded in TPA (252 by TPA vs. 289 by IL2+DSP30, *p* = 0.046). No significant differences in the type of abnormalities identified in each culture were found (*p* = 0.409); structural aberrations were more frequently detected than numerical ones (63.5% [160/252] using TPA and 69.6% [201/289] using IL2+DSP30). In addition, a median of one alteration (range: 1–13) was found among those abnormal karyotypes found by both mitogens. As for the abnormalities identified, highly similar profiles of alterations were obtained (Figure 2A). The most frequently detected abnormalities, found in at least five patients using one of the CBA cultures, included mainly alterations previously well-defined in CLL. Indeed, no significant differences were obtained for their detection rates by TPA and IL2+DSP30 cultures, respectively, as follows: +12 (n = 36 vs. 38, *p* = 0.801), del(13q) (n = 14 vs. 29, *p* = 0.017), del(11q) (n = 11 vs. 15, *p* = 0.421), structural aberrations at 17p (n = 13 vs. 12, including del(17p), i(17q) and unbalanced rearrangements with loss of 17p, *p* = 0.838), 18q21 translocations (n = 10 vs. 13, *p* = 0.522), 14q32 translocations (n = 9 vs. 10, *p* = 0.815), 13q14 translocations (n = 8 vs. 9, *p* = 0.805), and del(6q) (n = 6 vs. 5, *p* = 0.761). As expected, balanced translocations were found in a low proportion of patients (n = 24 [9.4%] using TPA and n = 35 [13.8%] using IL2 + DSP30, *p* = 0.128), with t(14;18)(q32;q21) being the most frequent (n = 8 vs. 9, *p* = 0.805). 

### 3.3. Complex Karyotype Detection

A similar CK detection rate was achieved using both mitogens, with CK being identified in a total of 26 patients (10.2%) using TPA and 29 patients (11.4%) using IL2+DSP30 (*p* = 0.775). However, when the information from both cultures was combined, the global detection rate of CK increased to 12.9% (n = 33). Among them, 19 were classified as high-CK. Of note, only 22/33 (66.7%) were concordantly detected using both methods, which also allowed us to stratify them as low-/intermediate-CK or high-CK in the same way (9 and 13 patients, respectively). Most of the discordant complex cases were only identified using IL2+DSP30 (7/11, 64%; 2.7% of the whole cohort), hence only four patients with CK from the whole cohort (1.6%) would remain unidentified without the assessment of TPA-based cultures. In addition, the number of abnormalities present in the CK did not significantly impact the concordance between cultures (discordant detection in 5/14 (35.7%) low-/intermediate-CK vs. 6/19 (31.6%) high-CK, *p* = 0.172). As for the cause of the discrepancy, in seven patients, it was due to the absence of metaphases or normal results obtained using the other mitogen, while in four patients, an abnormal karyotype (1–2 abnormalities) was detected in the other culture (Table 2). Interestingly, in three of the latter, the CK detected was a clonal evolution from the non-complex clone that was expanded only in the other culture. 

Even though CK groups found using TPA or IL2+DSP30 did not include the same patients, both displayed a median of 5 abnormalities (range: 3–13). As expected, CK included several structural and numerical aberrations distributed along the genome. Regarding the aforementioned recurrent abnormalities in the present cohort, the only ones found in a significantly higher proportion in CK compared with the non-CK patients were del(11q) (2.2% vs. 23.1% using TPA, *p* < 0.001 and 2.7% vs. 31.0%, using IL2+DSP30, *p* < 0.001) and trisomy 12 when assessed using TPA (12.2% vs. 30.8%, *p* = 0.017) (Figure 2B). In addition, CKs were significantly enriched in unbalanced structural abnormalities and monosomies. Detailed karyotypes from the 33 patients carrying a CK in at least one of the cultures are shown in Table 3.

### 3.4. Impact of Patients’ Status on the CBA Result

According to disease status, the abnormality detection rate obtained in patients with an active disease was significantly higher both in TPA- (35.2% vs. 57.8%, *p* = 0.007) and IL2+DSP30-based cultures (46.7% vs. 64.4%, *p* = 0.034) compared to those with a stable disease. Expectedly, CK was most frequently detected when a CBA study was performed in active disease patients using both methods (TPA CK detection: 5.2% vs. 33.3%, *p* < 0.001; and CK using IL2+DSP30: 6.2% vs. 35.6%, *p* < 0.001). Of note, both statuses showed similar concordance between cultures in terms of the detection of abnormal karyotypes (κ index: 0.581 and 0.650, for stable and active cases, respectively) and CK detection (κ index: 0.735 and 0.754, respectively).

### 3.5. Clinical Characteristics of Patients with CK Detected using TPA and/or IL2+DSP30

When combining results from both cultures, the CK group showed a significantly higher incidence of poor prognosis genetic features than those with non-CK, as follows: *ATM* and *TP53* deletion by FISH (33% each) and unmutated IGHV (83%) (Table 1). In addition, even though the CBA analysis of these patients and those with non-CK was performed mostly at diagnosis, the CK group showed significant enrichment in patients with active disease (55% vs. 12%, *p* < 0.001) and a shorter median TTFT (15 months vs. not reached, *p* < 0.001). The clinical and genetic characteristics of those patients with CK only identified using TPA or IL2+DSP30 (4 and 7 patients, respectively) were heterogeneous, and none of the features allowed to discriminate them from those 22 concordantly detected. However, it is important to note that the small cohort size precluded performing statistical analyses.

Regarding the prognostic impact of CK stratification, the CK group defined by both methods showed similar TTFT, which was significantly shorter than the TTFT observed in the non-CK group using both TPA (18 months vs. not reached, *p* < 0.001) and IL2+DSP30 (18 months vs. not reached, *p* < 0.001). However, CBA cultures using the IL2+DSP30 protocol displayed better accuracy than those with TPA for predicting TTFT (C-index: 0.580 by TPA vs. 0.605 by IL2+DSP30). As only seven high-CK patients detected using TPA and ten detected using IL2+DSP30 were treatment-naïve at the time of this study, the prognostic impact of high-CK could not be evaluated. On the other hand, non-complex abnormal karyotypes were associated with a shorter TTFT than normal cases only when identified using TPA (Figure 3).

## 4. Discussion

In recent years, the development of novel therapeutic strategies has reinvigorated the interest in the prognostic and potentially predictive value of CK in CLL. In this context, current ERIC recommendations consider CBA a standardized and feasible method for cytogenetic characterization [9]. Hence, they suggest the parallel analysis of two cultures using TPA and IL2+DSP30 as mitogens. Although the good performance of the IL2 and DSP30 combination has been largely demonstrated, evidence of the real benefit of combining cultures in the same patients is scarce [6,12,14,18]. In addition, none of the few studies that have compared both mitogens in the same cohort have specifically addressed which is the best practice to maximize CK detection in CLL [11,15,16]. Herein, we described the largest study evaluating the performance of the assessment of parallel TPA and IL2+DSP30 cultures in a single institution for CK detection in CLL. It should be noted that in contrast to previous studies, our series was analyzed in a single center with homogeneous sample handling, processing, and analysis.

Regarding the global performance, our study confirmed that CBA of IL2+DSP30 cultures achieved higher alteration detection rates than those of TPA-based cultures. In accordance with previous studies, IL2+DSP30 stimulation also allowed us to obtain a higher proportion of abnormal metaphases in the majority of the patients showing alterations in both cultures [11,15]. Moreover, both methods identified a similar distribution of aberrant chromosomal regions, mostly including abnormalities that were previously well-defined in CLL (e.g., 11q-, +12, 13q-, IGH rearrangements, 17p aberrations) among a broad spectrum of abnormalities along the whole genome. The percentage of abnormal cases found using IL2+DSP30 obtained herein (50%) is lower than the approximately 80% described in the initial studies, which allowed the introduction of this combination in CLL routine practice [12,14]. Nonetheless, it is important to note that most of the large CBA studies using IL2+DSP30 in unselected CLL cohorts, or in patients at diagnosis, also showed similar taxes of abnormal karyotypes, ranging from 52 to 68% of the whole cohort [6,11,15,18]. Hence, this variance could be associated with intrinsic differences in the clinical characteristics of the assessed cohorts rather than with technical issues. In accordance, the percentage of aberrations detected in the present study could be influenced by the cohort composition, which was highly enriched in newly diagnosed CLL patients at the Binet A stage with indolent disease and including patients with CLL-like monoclonal B cell lymphocytosis.

When genomic complexity was specifically addressed, both mitogens identified around 11% of patients displaying a CK, which is a frequency expected for a treatment-naïve CLL cohort [6,19]. This finding is also consistent with the two previous comparative analyses in unselected CLL patients, which described CK detection rates ranging from 11 to 17%, and neither identified significant differences between TPA and IL+DSP30 cultures [11,15]. However, considering each individual patient, detection was not fully concordant, and nearly one-third of the patients with CK could only be identified in one of the cultures. The best CK detection rate was obtained by combining the two methods. Notably, most of the discordant complex cases were only identified in the CBA with IL2+DSP30. The main causes of the discrepancy, found in seven patients, were the detection of a normal karyotype or the absence of metaphases in the other culture, suggesting that the mitogen failed to stimulate the malignant clones in those specific cultures. In contrast, the stratification based on the number of aberrations was concordant in 96% of the 91 patients showing abnormal karyotypes using both mitogens. Specifically, all the concordant CKs were equally classified as low-/intermediate-CK or high-CK, and only four patients with CKs displayed an abnormal non-complex karyotype in the parallel culture. Interestingly, three of the latter showed clonal evolution, and some of the abnormal related clones were differentially stimulated by the mitogens. The limited number of patients showing CK hampered the evaluation of the potential impact of clinical or genetic characteristics in the differential growth of the abnormal clone with CK in both conditions.

On the other hand, we evaluated whether the disease status of the patients at the time of CBA could influence concordance in the detection of abnormalities. As expected, those patients who presented with an active disease showed a higher frequency of abnormal karyotypes and CKs than stable patients. In this regard, it could be hypothesized that the malignant clones from those patients with active disease were more proliferative in vitro, independently of the mitogen used. Nonetheless, no differences in the agreement among cultures were observed between stable and active cases neither in terms of detection of abnormalities nor CK identification.

To date, it has been extensively demonstrated that CK is associated with an aggressive clinical outcome in patients treated with chemoimmunotherapy, with those with a high CK (≥5 abnormalities) showing the worst prognosis, independently of other factors [6]. In addition, mounting studies have suggested its prognostic role in the era of targeted therapies [20,21,22]. The prognostic evaluation of the CK was not the main aim of this study, as the size of the cohort did not allow for dividing the group into low-/intermediate-CK and high-CK for the survival analyses. Despite this limitation, we performed a comparative analysis of the impact of the CKs identified using each mitogen on TTFT. To the best of our knowledge, this is the first publication comparing the prognostic impact of CKs identified in the two parallel cell cultures from the same cohort of patients. Our results demonstrated that although patients included in each CK group were not fully equivalent, complexity was significantly associated with a poorer evolution when identified using either TPA or IL2+DSP30. Nonetheless, better accuracy for TTFT prediction was achieved with IL2+DSP30. Interestingly, patients with a non-complex abnormal karyotype only showed a dismal evolution compared with those with a normal result if assessed with TPA. This observation was most likely a consequence of the better specificity of the IL2+DSP30 combination to stimulate tumor cells than the poorer outcome of the subgroup of patients identified using TPA. While the group with normal karyotypes defined by IL2+DSP30 only contained 4% of patients with abnormal results using TPA, a considerable proportion of altered cases were masked in the normal group defined by TPA (up to 20% of the normal cases being aberrant in the parallel IL2+DSP30 culture). Hence, the group showing one or two abnormalities using TPA could be enriched in patients with more active disease than those who remained undetected and were masked in the normal group, as their tumor cells showed a higher proliferation capacity despite not being highly stimulated by the mitogen. Nonetheless, additional supplementary analyses should be performed to achieve solid conclusions. In line with our results, the largest study of the impact of genomic complexity detected using CBA to date described a worse distinction between low-CK/intermediate-CK and high-CK among cases analyzed with TPA [9]. Hence, these results suggested that the TPA protocol may fail to reveal the full spectrum of chromosomal abnormalities within the CLL clone, potentially underestimating some patients with an abnormal karyotype or CK.

## 5. Conclusions

In summary, in line with current ERIC recommendations, the present study shows that CBA combining results from cultures using TPA and IL2+DSP30 as mitogens is the best option to detect CK in CLL patients [9]. Nevertheless, it is important to note that the IL2+DSP30 protocol achieved superior performance than TPA. Indeed, considering the whole cohort, the percentage of patients with CK that could be identified when adding information from TPA cultures to the result obtained with IL2+DSP30 was low. Therefore, as the expense of money and time in harvesting and analyzing two cytogenetic cultures could be limiting in some routine laboratories, the use of IL2+DSP30 should be prioritized above TPA. Based on our results, in which the main cause of discrepancy was culture failure or no stimulation of the abnormal CLL clone (normal karyotype masking the altered clone), our proposal to optimize CBA in routine practice is to set up both cultures. However, we suggest assessing only CBA from IL2+DSP30 as the first step, and then, if no abnormalities could be identified, analyzing metaphases from the TPA-based culture as the second step.

In addition, the aforementioned limitations of the in vitro stimulation of the malignant CLL clones would potentially be overcome by the introduction of optical genome mapping (OGM) in current cytogenetic practices in the near future. OGM is a high-resolution technique that relies on the analysis of long DNA molecules (≥250 Kb) that are fluorescently labeled in specific sequences. Mapping the labeled molecules to reference the genome allows us to see gains or losses, copy number changes, and balanced structural events, all in a single test. We and others have demonstrated that OGM is a valuable tool for CK assessment in CLL [23], and it could also complement or even potentially replace FISH for the analysis of the abnormalities included in Döhner’s hierarchical model [24]. Nonetheless, standard criteria to define genomic complexity using OGM and additional studies of the clinical significance of OGM findings are mandatory for its implementation in the routine cytogenetic assessment in CLL patients.

## Figures and Tables

**Figure 1 cancers-16-02258-f001:**
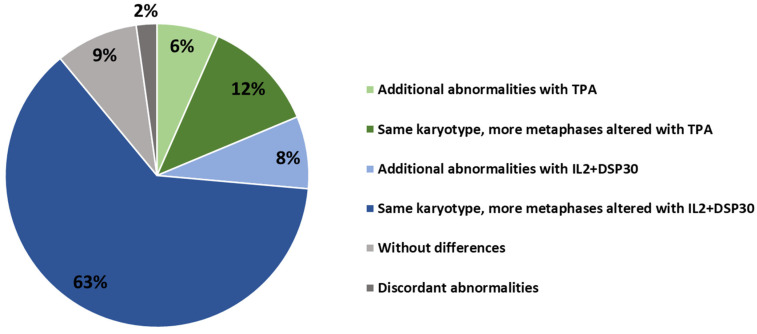
Comparison of TPA and IL2+DSP30 culture results in patients with altered karyotypes found using both methods (n = 91).

**Figure 2 cancers-16-02258-f002:**
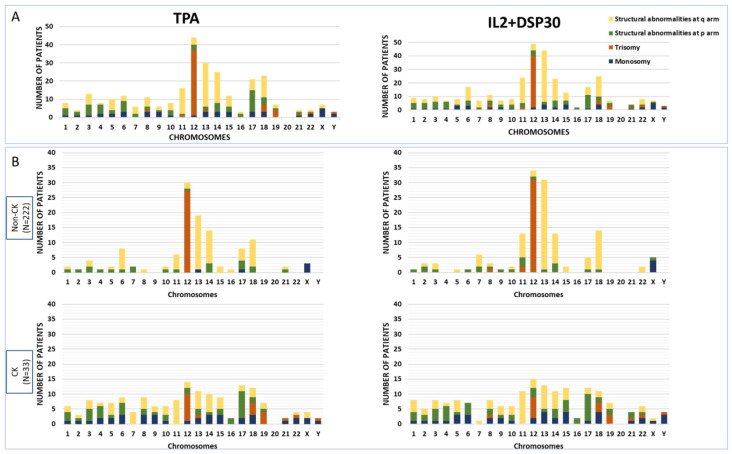
(**A**) Distribution of chromosomal abnormalities detected using TPA and IL2+DSP30 in the global cohort. (**B**) Distribution of chromosomal abnormalities detected using TPA and IL2+DSP30 in non-CK and CK groups. Patients were considered complex if a CK was detected in at least one of the cytogenetic cultures.

**Figure 3 cancers-16-02258-f003:**
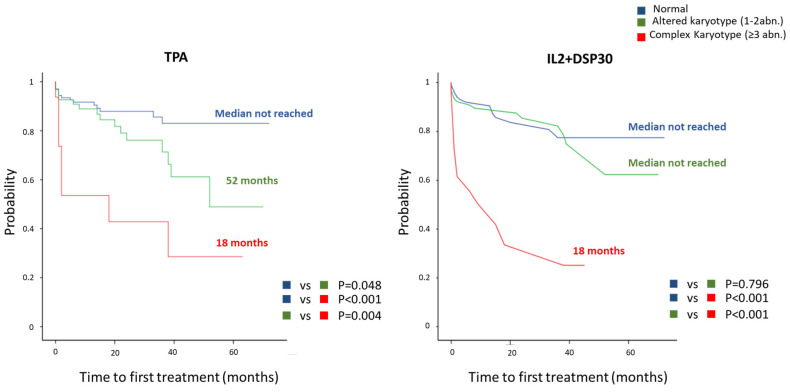
Kaplan-Meier plots of time to first treatment for TPA and IL2+DSP30 cultures in patients with normal karyotypes, karyotypes showing 1–2 cytogenetic abnormalities, or with complex karyotypes.

**Table 1 cancers-16-02258-t001:** Clinical and biological characteristics of the studied cohort. Patients were divided into non-CK and CK groups, with those with a CK detected in at least one of the cytogenetic cultures being classified as complex.

	Non-CK Group (n = 222)	CK Group (n = 33)	*p*-Value
**Gender**			
Men	122/222 (55%)	18/33 (55%)	1.000
**Median age at diagnosis**	69 years [30–99]	70 years [37–90]	0.279
**Stage at diagnosis**			
MBL	73/222 (33%)	7/33 (21%)	0.229
CLL	149/222 (67%)	26/33 (79%)	
Binet A	126/147 (86%)	20/26 (77%)	0.383
Binet B/C	21/147 (14%)	6/26 (23%)	
**Genomic alterations**			
del(13)(q14)	121/222 (55%)	19/33 (58%)	0.852
Trisomy 12	30/222 (14%)	8/33 (24%)	0.118
del(11)(q22q23)	12/222 (5%)	11/33 (33%)	<0.001
del(17)(p13)	10/222 (5%)	11/33 (33%)	<0.001
*TP53* Mutation (n = 134)	6/115 (5%)	5/19 (26%)	<0.001
**unmutated IGHV (n = 135)**	43/112 (38%)	19/23 (83%)	<0.001
**Median follow-up**	22 months [0–72]	23 months [0–65]	0.981
**Time from diagnosis to cytogenetic study**	3 months [0–430]	5 months [0–261]	0.576
**Treatment**			
Treated patients	44/222 (20%)	23/33 (70%)	<0.001
**Median time to first treatment**	NR [0–399]	15 months [0–399]	<0.001
**Patient status at the time of the cytogenetic study**			
Stable disease	195/222 (88%)	15/33 (45%)	<0.001
Active disease	27/222 (12%)	18/33 (55%)	

NR: not reached.

**Table 2 cancers-16-02258-t002:** Classification of patients according to the number of aberrations detected by CBA using TPA and IL2+DSP30 as mitogens.

		CBA Result Obtained with TPA					
		Normal Karyotype	Abnormal Non-CK (1–2 abn)	Low/Intermediate-CK (3–4 abn)	High-CK (≥5 abn)	No Metaphases	Total
**CBA result obtained with IL2+DSP30**	**Normal Karyotype**	103	5	0	0	8	116 (45.5%)
	**Abnormal non-CK (1–2 abn)**	26	65	1	1	5	98 (38.4%)
	**Low/intermediate-CK ** **(3–4 abn)**	0	1	9	0	2	12 (4.7%)
	**High-CK (≥5 abn)**	1	1	0	13	2	17 (6.7%)
	**No metaphases**	6	2	1	1	2	12 (4.7%)
	**Total**	136 (53.3%)	74 (29.0%)	11 (4.3%)	15 (5.9%)	19 (7.5%)	

**Table 3 cancers-16-02258-t003:** Detailed karyotypes and FISH results detected in the 33 patients showing a CK in the TPA and/or IL2+DPS30 culture. Patients were listed taking into account the concordance between methods in terms of detection of low-/intermediate-CK (3–4 abnormalities) or high-CK (≥5 abnormalities).

ID	Karyotype by TPA	Karyotype by IL2+DSP30	FISH
CONCORDANT KARYOTYPES			
**Low-/Intermediate-CK using both mitogens**			
**1**	46,XY,t(13;13)(q31;q14)[10]/46,XY,t(3;5)(q26;q13),t(13;13)(q31;q14), del(15)(q23),add(17)(p12)[4]/46,XY [6]	46,XY,t(13;13)(q31;q14)[10]/ 46,XY,t(3;5)(q26;q13),t(13;13)(q31;q14),del(15)(q23),add(17)(p12)[4]/ 46,XY [6]	Normal FISH
**2**	50,XY,+Y,+12,+18,+19[20]	50,XY,+Y,+12,+18,+19[6]/46,XY[14]	+12
**3**	49,XY,+12,+18,+19[6]/46,XY[14]	49,XY,+12,+18,+19[15]/46,XY[5]	13q- and +12
**4**	46,XX,del(11)(q21q24)[4]/46,X,del(X)(q22qter),add(4)(p16), del(10)(q24qter),del(11)(q21q24)[16]	46,XX,del(11)(q21q24)[4]/46,X,del(X)(q22qter),add(4)(p16), del(10)(q24qter),del(11)(q21q24)[20]	13q- and 11q-
**5**	47,XY,der(6)add(6)(p25),del(6)(q21),del(11)(q14q23),+mar[13]/46,XY[7]	47,XY,der(6)add(6)(p25),del(6)(q21),del(11)(q24q23),+mar[20]	13q- and 11q-
**6**	45,XX,add(11)(q25),der(15)t(15;18)(p11q11),-18[12]/46,XX[8]	45,XX,add(11)(q25),der(15)t(15;18)(p11q11),-18[19]/46,XX[1]	13q-
**7**	44,XX,der(4)t(4;15)(p16;q21),tas(6;8)(p25;q24), -15,dic(17;18)(p11;p11)[12]/46,XX[5]	44,XX,der(4)t(4;15)(p16;q21),tas(6;8)(p25;q24),-15, dic(17;18)(p11;p11)[20]	17p-
**8**	49,XY,+12,+18,+19[7]/49,XY,+12,del(13)(q14q22),+18,+19[9]/46,XY[4]	49,XY,+12,+18,+19[16]/49,XY,+12,del(13)(q14q22),+18,+19[3]/46,XY[1]	13q- and +12
**9**	46,XX,t(3;6)(q21;p24),t(5;9)(q13;p23),add(17)(p13)[5]/46,XX[10]	46,XX,t(3;6)(q21;p24),t(5;9)(q13;p23),add(17)(p13)[5]/46,XX[10]	Normal FISH
**High-CK using both mitogens**			
**10**	46,XY,del(11)(q21q24),ins(12;?)(q12;?),del(14)(q22q31), add(19)(q13)[7]/46,XY,del(11)(q21q24),ins(12;?)(q12;?), del(14)(q22q31),t(18;22)(q21;q22),add(19)(q13)[13]	46,XY,del(11)(q21q24),ins(12;?)(q12;?),del(14)(q22q31), add(19)(q13)[5]/46,XY,del(11)(q21q24),ins(12;?)(q12;?), del(14)(q22q31),t(18;22)(q21;q22),add(19)(q13)[15]	11q-
**11**	46,XX,-1,del(3)(p13),-5,der(10)t(1;10)(?;q22),del(11)(q23), +mar1,+mar2[20]	46,XX,-1,del(3)(p13),-5,der(10)t(1;10)(?;q22),del(11)(q23), +mar1,+mar2[20]	11q-
**12**	46–50,XY,add(1)(p36),add(2)(q34),+add(2)(q34),del(3)(p11p22), del(8)(p21),add(9)(q34),add(10)(p11),-12,-13,del(13)(q13), +21,+22,+3mar[cp10]	46–50,XY,add(1)(p36),add(2)(q34),+add(2)(q34),del(3)(p11p22), del(8)(p21),add(9)(q34),add(10)(p11),-12,-13,del(13)(q13), +21,+22,+3mar[cp10]	13q- and 17p-
**13**	46,XX,i(8)(q10),der(18)t(15;18)(q23;p11)[17]/44,XX,i(8)(q10),-9,add(10)(p15),-15,add(16)(p12),der(18)t(15;18)(q23;p11)[1]/ 45,XX,-4,-10,-14,add(14)(q32),der(18)t(15;18)(q23;p11),+2mar[2]	46,XX,i(8)(q10),der(18)t(15;18)(q23;p11)[11]/44,XX,I(8)(q10), -9,add(10)(p15),-15,add(16)(p12),der(18)t(15;18)(q23;p11)[4]/ 46,XX,i(8)(q10),-12,der(18)t(15;18)(q23;p11),+mar[2]/ 46,XX,t(3;9)(p14;q34),t(14;15)(q35;q22)[3]	13q-
**14**	44,XX,-6,del(12)(p12),-15,add(17)(p11),der(19)t(6;19)(q12;p13)[2]/ 46,XX[18]	45,XX,-6,add(17)(p11),der(19)t(6;19)(q12;p13)[7]/44,XX,-6,del(12)(p12), -15,add(17)(p11),der(19)t(6;19)(q12;p13)[6]/44,XX,-6,add(12)(q24), -15,add(17)(p11),der(19)t(6;19)(q12;p13)[2]/46,XX[5]	17p-
**15**	46,XX,add(15)(q26)[6]/42–43,XX,add(4)(p11),del(5)(q14q32), -6,add(13)(p11),-14,add(15)(q26),add(17)(p12),-18[cp8]/46,XX[6]	42–43,XX,add(4)(p11),del(5)(q14q32),-6,add(13)(p11), -14,add(15)(q26),add(17)(p12),-18[cp9]/ 42,XX,add(4)(p11),-5,-6,add(7)(q32),-14,add(15)(q26),add(17)(p12), -18[6]/46,XX[5]	17p-
**16**	46,X,-Y,del(3)(p14),t(8;13)(q22;q14),+12[15]/ 45,X,-Y,der(5)t(1;5)(p22;p15),+12,-14,der(17)t(14;17)(q11;p11)[4]/ 46,XY[1]	46,X,-Y,del(3)(p14),t(8;13)(q22;q14),+12[12]/47,X, -Y,del(3)(p14),t(8;13)(q22;q14),+12,der(12)t(12;?)(p11;?),+mar[6]/ 45,X,-Y,der(5)t(1;5)(p22;p15),+12,-14,der(17)t(14;17)(q11;p11)[2]	13q-,+12 and 17p-
**17**	45,XY,-4,+12,-13,der(16)t(4;16)(q11;p13),add(17)(p12), der(21)t(4;21)(p11;p11)[20]	45,XY,-4,+12,-13,der(16)t(4;16)(q11;p13),add(17)(p12), der(21)t(4;21)(p11;p11)[10]	13q- +12 and 17p-
**18**	45,X,-Y,-3,add(9)(q34),add(12)(p11),add(14)(p11), -17,+marx2[20]	45,X,-Y,-3,add(9)(q34),add(12)(p11),add(14)(p11),-17,+marx2[7]/ 46,XY[13]	17p-
**19**	46,XX,add(2)(p24),del(11)(q21q23)[6]/45,sl,-5,-6,-8,-18,-21, -22,+5mar[2]/90,sdlx2[2]/46,XX[10]	46,XX,add(2)(p24),del(11)(q21q23)[8]/45,sl,-5,-6,-8,-18,-21, -22,+5mar[9]/90,sdlx2[1]/46,XX[2]	11q-
**20**	47,XX,+12[3]/48,XX,+8,del(10)(q23),+12,del(14)(q22q31), add(19)(q13)[2]/46,XY[15]	46,XX,del(14)(q22q31)[9]/ 47,XX,+12[1]/48,xx,+8,del(10)(q23),+12,del(14)(q22;q31), add(19)(q13)[7]/46,XY[3]	Normal FISH
**21**	44,X,-X,del(13)(q14q22),add(15)(p11),add(17)(p11),-22[3]/46,XX[17]	44,X,-X,del(13)(q14q22),add(15)(p11),add(17)(p11),-22[12]/46,XX[8]	13q- and 17p-
**22**	45,XX,add(1)(q44),-2,-8,-9,del(11)(q21q23),add(12)(q24), add(14)(q32),add(17)(p13),+mar1,+mar2[6]/46,XX[7]	45,XX,add(1)(q44),-2,-8,-9,del(11)(q21q23),add(12)(q24), add(14)(q32),add(17)(p13),+mar1,+mar2[20]	11q-
**DISCORDANT KARYOTYPES**			
**High-CK only using IL2+DSP30**			
**23**	46,XY[20]	46,XY,add(11)(q23)[6]/44,xy,add(11)(q23),der(14)t(14;18)(p11;q11), -15,-18,der(22)t(15;22)(p11;q15)[9]/45,XY,add(11)(q23), der(14)t(14;18)(p11;q11),-15,-18,der(22)t(15;22)(p11;q15),+mar[5]	13q- and 11q-
**24**	47,XX,del(11)(q22q23),+12[4]/46,XX[14]	47,XX,del(11)(q22q23),+12[4]/47,XX,del(11)(q22q23),+12, -13,add(15)(p11),add(19)(p13)[1]/45,XX,i(8)(q10), del(11)(q22q23), +12,-13,add(15)(p11),add(19)(p13)[15]	13q- and 11q-
**25**	No metaphases	46,XY,add(2)(p23),-10,del(13)(q14q34),+mar1[6]/ 46,XY,add(2)(p23),-10,del(13)(q14q34),i(17)(q10),+mar2[2]/46,XY[12]	13q-,11q- and 17p-
**26**	No metaphases	45,X,-Y,inv(3)(q21q26),add(6)(p24),add(15)(p11)[9]/45,X,-Y, inv(3)(q21q26),del(5)(q31q33),add(6)(p24),add(15)(p11)[8]/ 45,X,-Y,inv(3)(q21q26),add(6)(p24),add(14)(p11),add(15)(p11)[3]	13q-
**High-CK only using TPA**			
**27**	44,XX,-8,-9,add(13)(q34),der(17)t(8;17)(q11,p11)[4]/ 43,X,-X,der(3),ins(?;3)(?;p14),del(3)(p13),-8,-9,add(13)(q34), der(17)t(8;17)(q11;p11)[11]/46,XX,der(13)t(13;?)(p11;?)[4]/46,XX[3]	No Metaphases	13q- and 17p-
**28**	47,XY,+12[9]/47,XY,del(X)(q25),add(5)(q31),add(8)(q24),+12, del(14)(q22q32)[4]/46,XY[7]	47,XY,+12[16]/46,XY[4]	+12
**Low-/Intermediate-CK only using IL2+DSP30**			
**29**	No metaphases	46,XY,del(11)(q23q25)[12]/46,XY,t(1;2)(q32;q37),del(4)(p11), del(6)(q16),del(11)(q23q25)[cp6]/46,XY[2]	13q- and 11q-
**30**	No metaphases	46,XX,del(11)(q14q24),del(13)(q14q22)[5]/45,XX,del(11)(q14q24), -13,der(21)t(13;21)(q11;p11),del(13)(q14q22)[13]/46,XX[2]	13q- and 11q-
**31**	46,XY,add(17)(p13)[10]/46,XY,del(13)(q14q22), add(17)(p13)[3]/46,XY[7]	46,XY,add(17)(p13)[3]/46,XY,del(13)(q14q22),add(17)(p13)[4]/ 46,XY,dup(1)(q24q44),del(13)(q14q22),add(17)(p13)[2]/46,XY[7]	13q-
**Low-/intermediate-CK only using TPA**			
**32**	49,XX,+12,+18,+19[12]/46,XX[8]	No metaphases	13q- and +12
**33**	46,XY,-17,+mar[7]/46,XY,add(6)(p25),-17,+mar[2]/46,XY,dic(17;18)(q10;q10)[1]/46,XY[10]	46,XY,dic(17;18)(q10;q10)[1]/46,XY[19]	17p-

Abbreviations: 13q: deletion in 13q14; 11q-: *ATM* deletion (11q23); +12: trisomy 12; 17p-: *TP53* deletion (17p13).

## Data Availability

Detailed chromosome banding analyses from complex cases are provided in Table 2. Additional data of this study are available from the corresponding authors upon reasonable request.

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
