# Peer review of "Complex Karyotype Detection in Chronic Lymphocytic Leukemia: A Comparison of Parallel Cytogenetic Cultures Using TPA and IL2+DSP30 from a Single Center"

_cancers, 2024, doi:10.3390/cancers16122258_

Round 1
Reviewer 1 Report
Comments and Suggestions for Authors
In this study, the authors evaluated the performance of two parallel cultures using TPA and IL2+DSP30 mitogens in a CLL cohort of consecutively received unselected patients in a single center. The authors concluded that the analysis of two parallel cultures was the best option to detect CK in CLL, and IL2+DSP30 could be prioritized above TPA to optimize cytogenetic assessment in clinical practice.
Comments:
The reviewer has some concerns as follows:
1. The authors have fully explained the motivation for this study and the relevant research literature. However, one of the major concerns about this study is its similarity to a previous literature (Puiggros et al., Optical Genome Mapping: A Promising New Tool to Assess Genomic Complexity in Chronic Lymphocytic Leukemia (CLL). Cancers (Basel). 2022 Jul 11;14(14):3376.). The two have similar themes, research methods and results, and even the patients are recruited from a same medical center and share the same IRB number (Hospital del Mar Ethics Committee 2017/7565/I), although the recruited number of patients included in this study is larger. The authors should explain whether there is an academic ethics issue, and describe in detail the difference between these two studies and whether they share the same patient data.
2. In the Methods, please indicate which time and place (center) for recruiting patients.
3. The position of the legend for Figure 1 needs to be adjusted, please move it to the bottom of the figure.
4. In Table 3, it is confusing and unconvincing. Please reorganize this table.
5. In Figure 3, it seems unnecessary to frame the numbers on the x and y axes.
6. Overall, in the present state, the presented results cannot support the conclusions.
Author Response
Please see the attachmen

Reviewer 2 Report
Comments and Suggestions for Authors
Manuscript examined the effectiveness of mitogens in CLL patients for karyotyping and found that the combination of IL2+DSP30 was better in detecting clonal abnormalities. We detected a combination in 99% of cases. These results suggest practicing the combined use of both mitogens to increase diagnostic accuracy in clinical practice. Some comments are below for authors consideration
Introduction
• Provide a general description of the current CLL research landscape in this area, indicating the gaps to which this study contributes.
· Elaborate on the clinical implications: How the use of IL2+DSP30 would affect results, treatment, treatment outcomes, and management.
• Addressing limitations in the study, for example, potential bias that might derive from a single-center design or a specific cohort of patients.
• Words like "indolent course" and "aggressive behavior" must be clearly defined, probably through clinical examples.
• Define on first mention the term "complex karyotype (CK), " that is, what CK and high-CK stand for.
· It is necessary to spell out any initialisms, such as CLL, FISH, and CBA, when they are first mentioned in order to ensure that everyone is aware of what is being referred to.
Methods
• Provide sufficient details of the CBA and FISH test protocols for the readers to know what steps were taken in the sample preparation and result analysis to appreciate the rigor and reproducibility of the methods of this study.
•Refer to Table 1 for the relevant data to support the statements you make regarding the clinical characteristics. For instance, the findings in Table 1 reveal that the occurrence of del(11)(q22q23) and del(17)(p13) was markedly higher in individuals from the CK cohort compared to those from the non-CK cohort (P < 0.001).
· Provide specific details for the description of protocols; for instance, " peripheral blood cultures were performed according to standard protocols. One culture was stimulated with TPA, whereas the other was stimulated with IL2+ DSP30. Karyotypes were described according to the International System for Human Cytogenetic Nomenclature (ISCN 2020). A minimum of 20 metaphases of each sample was analyzed."
Results and Discussion
• The text exhibits a lack of cohesion, as it shifts abruptly between discussing results and making comparisons without a discernible transition. Specifically, the mention of success rates (lines 165-167) is followed immediately by a conversation on clonal abnormalities (lines 169-171) without any connective tissue.
· The text provides exact numbers; for example, the number of patients who turned out with abnormalities after every method was quoted as 39% in TPA and 50% in IL2+ DSP30, without a critical inquiry into the reasons for these differences. Other precise details about the clinical pictures or the patients' characteristics in which these differences manifested would help a great deal.
• The discussion of the complex karyotypes (CKs) identified by each method is relatively detailed, but the context of the actual importance of these CKs in a clinical scenario would help. Why is detecting CK important, and how might differences in detection impact patient management?
· Although the success rates in detecting clonal abnormalities were compared between TPA and IL2+DSP30, it is unclear why IL2+DSP30 might be more effective or under what circumstances TPA might still be preferable. In this context, there could be a better understanding of the practical implications of such findings that will be conveyed to the reader.
• Combination of Methods: It is noted that when both methods are used in combination, a detection rate of up to 99% can be achieved. However, much is not discussed in relation to the feasibility and practical application of these two methods in routine clinical practice. Are the implications of the cost, time, and logistics significant?
• Clinical Relevance: This section discusses what types of abnormalities are detected but does not consider the clinical relevance of such findings. For example, what is the actual impact of finding more abnormalities when performing IL2+DSP30 on patient outcomes or treatment decisions?.
· Underscore the importance of standardizing mitogen application protocols across samples in order to ensure consistency.
· More stringent control measures ought to be implemented to minimize the impact of extraneous variables on the effectiveness of mitogens.
Round 2
Reviewer 1 Report
Comments and Suggestions for Authors
This revised manuscript has a great improvement. No further comments.